# The association between bullous pemphigoid and cognitive outcomes in middle-aged and older adults: A systematic review and meta-analysis

Qi Zhou[1]*, Zhenrong Xiong[2], Dejiang Yang[3], Chongyu Xiong[1], Xinming Li[3]*

1 Department of Neurology, The First People's Hospital of Fuzhou, Fuzhou, Jiangxi, China, 2 Public Relations Department, The First People's Hospital of Fuzhou, Fuzhou, Jiangxi, China, 3 Department of Neurology, First Hospital of Nanchang, Nanchang, Jiangxi, China

* 17807026495@163.com (QZ); 108579933@qq.com (XL)

**Data Availability Statement:** All relevant data are within the paper and its Supporting Information files.

## Abstract

### Background

Bullous pemphigoid (BP) is a rare autoimmune skin condition that causes large fluid-filled blisters on the skin, especially in older adults. BP has been linked to various diseases and medications, but its association with cognitive outcomes is unclear.

### Methods

We conducted a systematic review and meta-analysis of studies investigating the association between BP and cognitive outcomes, such as all-cause dementia, Alzheimer's disease, and vascular dementia in middle-aged and older adults. We searched PubMed, Embase, and Web of Science databases for relevant studies published up to March 2023. We included studies that reported odds ratios (ORs) or hazard ratios (HRs) with 95% confidence intervals (CIs) for the association between BP and cognitive outcomes. We pooled the ORs, or HRs using random-effects models and performed subgroup and sensitivity analyses to explore potential sources of heterogeneity.

### Results

The study selection process identified 13 studies for inclusion in the analysis, 11 studied arms of which used a case-control design and 7 studied arms of which used a cohort design. The studies were conducted primarily in Europe, with a few from Asia and the United States. The meta-analysis found that BP was associated with higher odds of all-cause dementia in middle-aged and older participants in both cohort studies(HR = 1.41,95% CI: 1.20–1.66, P = 0.000) and case-control (OR = 4.25, 95% CI, 2.73–6.61; P = 0.000). The study found no significant publication bias in the included studies. The meta-regression analyses identified some subgroups associated with significantly reported odds ratios in case-control association analysis, including Europe, BP diagnosed based on clinical, histology, immunofluorescence, and both adjustment status of NO and YES.

**Funding:** The author(s) received no specific funding for this work.

**Competing interests:** The authors have declared that no competing interests exist.

## Conclusions

Our meta-analysis suggests that BP is associated with an increased risk of all-cause dementia in middle-aged and older adults. Further studies are needed to elucidate the underlying mechanisms and causal relationship between BP and cognitive outcomes.

## Introduction

Bullous pemphigoid (BP) is a chronic autoimmune blistering disease that affects the skin and mucous membranes. It is characterized by the presence of autoantibodies against two hemi-desmosomes proteins, BP180 and BP230, which are involved in the adhesion of epidermal cells to the basement membrane [1]. BP mainly affects elderly people, with a peak incidence between 70 and 80 years of age [2]. The clinical manifestations of BP include tense blisters, erosions, urticarial plaques and pruritus, which can have a significant impact on patients' quality of life [3].

BP is not a rare disease, as it has been estimated that its global prevalence ranges from 2.4 to 42.7 per 100,000 population, with higher rates in Europe and North America than in Asia and Africa [4]. The incidence of BP has been increasing over the years, possibly due to the aging of the population and the improvement of diagnostic methods [4]. BP poses a considerable burden on the health care system and society, as it requires long-term treatment and monitoring, and it is associated with increased morbidity and mortality [5].

BP has been associated with various comorbidities, such as diabetes mellitus, cardiovascular diseases, malignancies, and thyroid disorders [6]. However, one of the most intriguing and controversial aspects of BP is its relationship with cognitive outcomes. Several studies have reported an increased prevalence of neurological disorders in BP patients, such as dementia, stroke, epilepsy, Parkinson's disease [7–9]. Moreover, some studies have suggested that BP may be a marker of cognitive decline or neurodegeneration, as BP patients have shown worse cognitive performance and higher mortality rates than controls [6,10,11]. However, other studies have failed to confirm these findings or have proposed alternative explanations for the association between BP and cognitive outcomes [12–14].

The relationship between BP and cognitive outcomes is not well understood and may involve multiple factors. Some possible mechanisms include the direct effect of autoantibodies on the central nervous system (CNS), the systemic inflammation induced by BP, the shared genetic susceptibility or environmental triggers between BP and neurological disorders, and the confounding effect of age or other comorbidities [15,16]. Therefore, it is important to confirm the association between BP and cognitive outcomes using rigorous methods and large samples, and to explore the potential mediators and moderators of this relationship.

Given the heterogeneity and inconsistency of the existing literature on this topic, a systematic review and meta-analysis is warranted to provide a comprehensive and objective assessment of the relationship between BP and cognitive outcomes. The aim of this study is to synthesize the available evidence on the relationship between BP and cognitive outcomes. This study will also explore potential sources of heterogeneity and bias among the included studies, such as methodological quality, diagnostic criteria, confounding factors, and publication bias. Our results can provide evidence for the early detection and intervention of cognitive decline in BP patients, as well as contribute to the understanding of the pathophysiology and mechanisms of BP and its neuro-cognitive effects.

## Methods

### Protocol and registration

This systematic review and meta-analysis aim to investigate the relationship between BP and cognitive outcomes, such as all-cause dementia, vascular dementia (VD), and Alzheimer's disease (AD). The protocol of this review was registered in the PROSPERO database and followed the PRISMA 2020 statement [17] (S1 Table) for reporting systematic reviews.

### Search strategy

We searched PubMed, Embase, and Web of Science from inception to March 2023 using the following terms: ("bullous pemphigoid" OR "pemphigoid") AND ("cognition" OR "cognitive" OR "dementia" OR "Alzheimer's" OR "vascular dementia" OR "mix dementia") AND ("middle-aged" OR "older adults") AND ("cohort" OR "case-control" OR "cross-sectional"), which is shown in S2 Table. We chose these terms based on the PICO framework, which consists of four components: Population, Intervention, Comparison, and Outcome. The population of interest was middle-aged and older adults with bullous pemphigoid. The intervention and comparison were the presence or absence of bullous pemphigoid, respectively. The outcome was the risk of cognitive impairment or dementia. We defined middle-aged and older adults as those aged 45 years or older, following the World Health Organization's definition [18]. In addition, we examined the bibliographies of pertinent articles to identify additional research.

### Inclusion criteria and exclusion criteria

We conducted a systematic review and meta-analysis of studies that examined the association between BP and cognitive outcomes in middle-aged and older adults. We aligned our inclusion and exclusion criteria with the PICO framework, which is a commonly used tool for framing systematic review research questions and developing literature search strategies. We included studies that: (1) were cohort, case-control, or cross-sectional in design; (2) reported quantitative data on the risk of cognitive outcomes in BP patients compared to a control group of non-bullous pemphigoid patients; (3) provided the effect size estimates HRs or ORs and their 95% CI; and (4) were published in English. We excluded studies that: (1) did not measure cognitive outcomes; (2) did not have a control group; (3) did not provide sufficient data for meta-analysis; (4) were case reports, reviews, meta-analyses, letters, editorials, or commentaries; or (5) had a quality score of less than 4 according to the Newcastle-Ottawa Scale (NOS). Two authors (X.L. and D.Y.) independently screened titles, abstracts, and full-text articles for eligibility. Any disagreement was resolved by a third author (Z.X.).

### Selection of studies and data extraction

Two reviewers (Q.Z. and C.X.) independently screened titles and abstracts, and then full texts of potentially eligible studies. Discrepancies were resolved by consensus or by a third reviewer. Data extraction was performed using a standardized form that included information on study characteristics, population characteristics, BP diagnosis, cognitive outcomes, and results. In case of missing data, we reached out to the corresponding author for additional information.

### Quality assessment of the included studies

We assessed the quality of the included studies using the NOS for cohort studies and case-control studies [19]. The NOS evaluates three domains: selection, comparability and outcome (or exposure for case-control studies). We rated each study as low, moderate, or high risk of bias based on the number and severity of the flaws in each domain. Data extraction and quality

assessment were conducted by two independent investigators (X.L. and D.Y.), and disagreements between them were resolved through negotiation with a third researcher (Z.X.).

## Data synthesis and analysis

We conducted a meta-analysis with a random-effects model to calculate the pooled effect sizes and 95% confidence intervals for each cognitive outcome between BP patients and non-BP patients(control group). We used ORs or HRs for binary outcomes. We evaluated heterogeneity with the I-squared statistic and investigated potential sources of heterogeneity with subgroup analyses and meta-regression based on study characteristics and population characteristics. We considered $I^2$ values of <25%, 25–50%, 51–75%, and >75% to indicate no, mild, moderate, and large heterogeneity, respectively [20]. We examined publication bias with funnel plots and Egger's test. We performed all analyses using Stata version 17 (Version 17.0; Stata SE Company LP, College Station, TX, USA).

## Results

### Study selection

We conducted a systematic search and identified 273 records. We screened the titles and abstracts of 60 articles and retrieved 43 full-text articles for further assessment (Fig 1). We

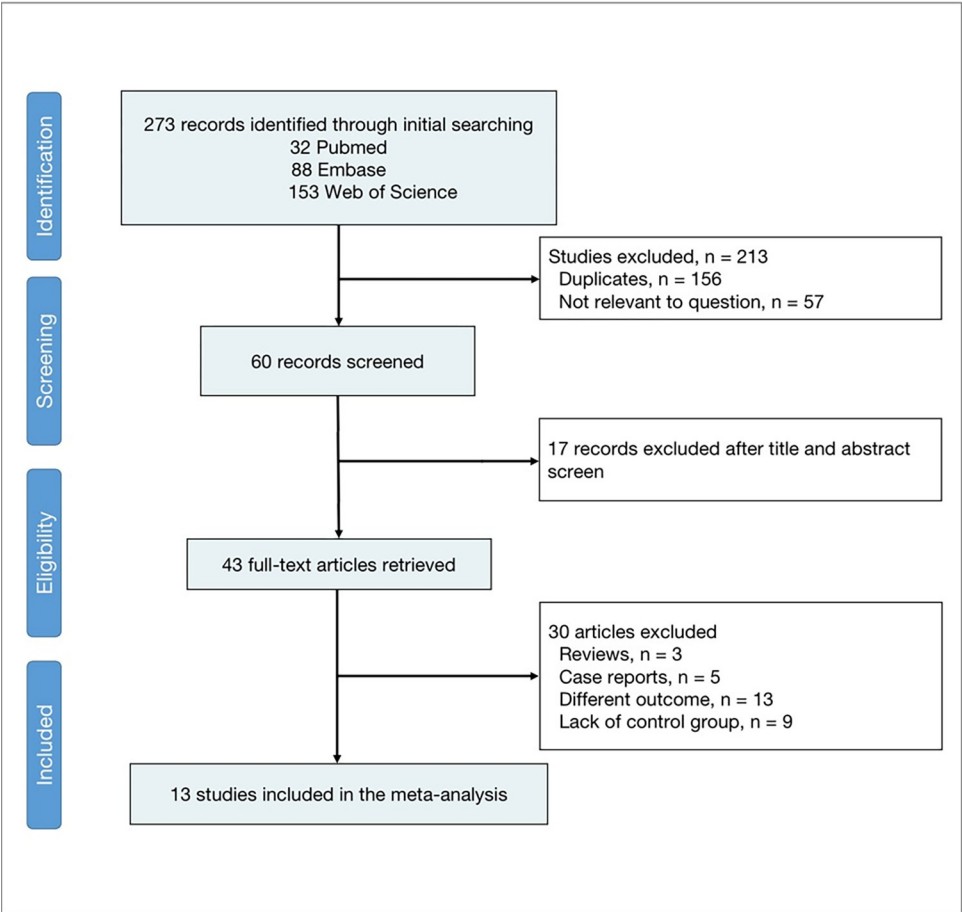

**Fig 1. Flowchart displaying the choice of study.**

excluded 30 studies based on the following criteria: review articles (n = 3), case reports (n = 5), different outcome measures (n = 13) and lack of an appropriate control group (n = 9). We included 13 studies in the meta-analysis that met our inclusion criteria:2 cohort studies [21,22], 10 case-control studies [9,12,14,23–29] and 1 study with both cohort and case-control study design [30].

## Characteristics of studies

Most of the studies (10 out of 13) used a case-control design, while 2 studies used a cohort design, and 1 study with both cohort and case-control study design. Most of the studies were conducted in Europe, with only 2 from Asia and 1 from the United States. Table 1 shows the main characteristics of each study. The diagnosis of cognitive outcomes was based on medical records in 10 studies and on the international coding of disease (ICD) in 3 studies. The diagnosis of BP was confirmed by clinical, histological, or immunofluorescence criteria in 8 studies and by ICD codes in 5 studies. All the case-control studies had moderate to high quality scores according to the NOS, with 4 scoring 8 stars (high quality) and 7 scoring 5–7 stars (moderate quality), and none of the studies had low quality scores (0–4 stars) (Table 2). All the cohort studies had high quality scores (Table 2). Most of the studies adjusted for potential confounders by matching or multivariate regression analysis. Among the all 18 studied arms from the 13 included articles, 11 case-control studied arms and 3 cohort studied arms examined the association between BP and all-cause dementia, 2 cohort studied arms examined the association between BP and AD dementia, and 2 cohort studied arms examined the association between BP and VD (seen in Table 1).

## Longitudinal cohort association between bullous pemphigoid and cognitive outcomes

We conducted a meta-analysis of three cohort studies that examined the association between BP and all-cause dementia in middle-aged and older participants (Fig 2A). The pooled HR of all-cause dementia among BP patients was 1.41 (95% CI: 1.20–1.66, P = 0.000). There was moderate heterogeneity between the studies ($I^2$ = 57.5%, P = 0.095). We also performed a sensitivity analysis to assess the influence of each study on the overall pooled HR for all-cause dementia. The results were consistent, and no single study altered the significance of the pooled HR (Fig 3A). The funnel plot did not show obvious publication bias (Fig 4A), and Egger's test did not indicate any small study effect (P = 0.470).

Systematic review finds only two cohort studies evaluated the longitudinal relationship between BP and AD and VD in middle-aged and older participants. Wotton et al [21] reported that BP patients had a significantly higher risk of AD (HR = 1.21: 1.06–1.38) and VD (HR = 1.65: 1.48–1.84) compared to control participants. Li et al [22] found no significant association between BP and AD (HR = 0.77,95% CI: 0.48–1.18) or VD (RR = 1.44,95% CI: 0.97–2.06) compared to control participants.

## Case-control association between bullous pemphigoid and cognitive outcomes

We pooled the estimates in case-control studies (Fig 2B) to examine the association between BP and all-cause dementia in middle-aged and older participants. We found that BP significantly increased the odds of all-cause dementia (pooling OR = 4.25, 95% CI, 2.73–6.61; P = 0.000). The studies had high heterogeneity ($I^2$ = 77.8%, P = 0.000). We also performed a sensitivity analysis to assess the influence of each study on the overall pooled OR for all-cause

**Table 1. Overview of studies on the association between bullous pemphigoid and cognitive outcomes in included studies.**

| Author, year | Baseline Study Years | Country or region | Population | No.Of Participants/Cases | Gender (% Female) | Study design | Diagnostic of bullous pemphigoid | Diagnosis of dementia | Follow-up (Years) | Mean Age (SD) | Adjustment for Potential Confounder | Type of Cognitive Dysfunction | Outcome |
|---|---|---|---|---|---|---|---|---|---|---|---|---|---|
| Papakonstaninou et al(2019) [1] | 2011-2015 | Germany | Clinic in Germany | 531/75 | 68 | Case-control | Based on the international coding of disease (ICD) | Medical Records | - | 80 | - | All-cause dementia | OR 9.9 (5.4-17.8) |
| Lin et al(2023) [2] | 2000-2013 | Taiwan | Longitudinal Health Insurance Database | - | 57.7 | Case-control | Based on the international coding of disease (ICD) | Based on the international coding of disease (ICD) | 13 | - | Monthly income, urbanization level, diabetes, hyperlipidemia, hypertension, and coronary heart disease | All-cause dementia | HR 1.31 (0.88-1.96) |
| Bienias et al (2019) [3] | 2000-2014 | Poland | Medical University of Warsaw | 368/50 | - | Case-control | Based on clinical, histology, immunofluorescence | Medical Records | - | 76.2±11.62 | Low-density lipoprotein cholesterol, and estimated glomerular filtration rate | All-cause dementia | OR 7.89 (2.99-20.85) |
| Wotton et al (2017A) [4] Wotton et al (2017B) [4] Wotton et al (2017C) [4] | 1998-2012 | United Kingdom | English national HES database | 1,833,827/1137 1,833,827/241 1,833,827/331 | 54.4 | Cohort study | Based on the international coding of disease (ICD) | Based on the international coding of disease (ICD) | 3.3 | 77.9 | Sex, age, time period in single calendar years, region and deprivation score | All-cause dementia Alzheimer disease dementia Vascular dementia | RR 1.53 (1.44-1.62) RR1.21 (1.06-1.38) RR1.65 (1.48-1.84) |
| Teixeira et al (2014) [5] | 1998-2010 | Portuguese | Database of patients at a Portuguese university hospital | 253/40 | - | Case-control | Based on clinical, histology, immunofluorescence | Medical Records | - | 79 | Age, gender and neurological disorders. | All-cause dementia | OR 5.25 (2.71-10.16) |
| Taghipour et al (2010) [6] | 2004-2008 | United Kingdom | Outpatient center for immunoblots diseases in England. | 231/14 | - | Case-control | Based on clinical, histology, immunofluorescence | Medical Records | - | - | Age and sex. | All-cause dementia | OR 7.9 (1.7-37.3) |
| MRCP et al(2015) [7] | 2004-2013 | Malaysia | Dermatology Unit | - | 55.8 | Case-control | Based on clinical features and immunofluorescence | Medical Records | - | - | Age and sex | All-cause dementia | OR 3.5 (1.2-10.3) |
| Li et al(2018A) [8] Li et al(2018B) [8] Li et al(2018C) [8] | 1964-2010 | Swedish | Swedish National Patient Register | - | 49.4 | Cohort study | Based on the international coding of disease (ICD) | Based on the international coding of disease (ICD) | - | 55.6 | Age, sex and comorbidities | All-cause dementia Alzheimer disease dementia Vascular dementia | SIR 1.26 (1.06-1.49) SIR 0.77 (0.48-1.18) SIR 1.44 (0.97-2.06) |

(Continued)

**Table 1.** (Continued)

| Author, year | Baseline Study Years | Country or region | Population | No.Of Participants/ Cases | Gender (% Female) | Study design | Diagnostic of bullous pemphigoid | Diagnosis of dementia | Follow-up (Years) | Mean Age (SD) | Adjustment for Potential Confounder | Type of Cognitive Dysfunction | Outcome |
|---|---|---|---|---|---|---|---|---|---|---|---|---|---|
| Foureur et al (2001) [9] | - | France | - | - | - | Case–control | Based on clinical, histology, immunofluorescence | Medical Records | - | - | - | All-cause dementia | OR 4.76 (1.97–11.51) |
| Casas-de-la-Asuncinet al(2014) [10] | 2002–2012 | Spanish | Cectronic medical record database in Córdoba, Spain. | 168/34 | - | Case–control | Based on clinical, histology, immunofluorescence | Medical Records | | - | Age and sex. | All-cause dementia | OR 5.52 (2.19–13.93) |
| Brick et al(2014A) [11] Brick et al(2014B) [11] | 1960–2009 | United States | Residents of Minnesota | 292/12 348/13 | 57 - | Case–control Cohort study | Based on clinical, histology, immunofluorescence | Medical Records | - | - | - | All-cause dementia All-cause dementia | OR 6.75 (2.08–21.92) HR 1.25 (0.61–2.55) |
| Langan et al(2011) [12] | 1996–2006 | United Kingdom | The Health Improvement Network | 4,337/143 | 62 | Case–control | Based on the international coding of disease (ICD) | Medical Records | - | 80 (23–102) | Charlson scores | All-cause dementia | OR 3.30 (2.30–4.60) |
| Bastuji-Garin et al (2011) [13] | 1997–2008, | France | National Health Insurance Research Database | 20,910/1420 | 45.2 | Case–control | Based on clinical, histology, immunofluorescence | Medical Records | - | 84.2 ±8.7 | Age, gender, centre and place residence matched | All-cause dementia | OR2.19 (1.24–3.87) |

**Table 2. Assessment of study quality.**

| Studies | Quality Indicators from Newcastle-Ottawa Scale | | | | | | | | | Sum |
|---|---|---|---|---|---|---|---|---|---|---|
| | 1 | 2 | 3 | 4 | 5a | 5b | 6 | 7 | 8 | |
| Papakonstaninou et al., (2019) [1] | 1 | 0 | 1 | 1 | 1 | 1 | 0 | 0 | 0 | 5 |
| Lin et al., (2019) [2] | 1 | 1 | 0 | 1 | 1 | 1 | 1 | 0 | 0 | 6 |
| Bienias et al., (2019) [3] | 1 | 1 | 1 | 1 | 1 | 1 | 1 | 1 | 0 | 8 |
| Wotton et al.,(2017) [4] | 1 | 1 | 1 | 1 | 1 | 1 | 1 | 1 | 0 | 8 |
| Teixeira et al.,(2014) [5] | 1 | 0 | 1 | 1 | 1 | 1 | 1 | 1 | 0 | 7 |
| Taghipour et al.,(2010) [6] | 1 | 1 | 1 | 1 | 1 | 1 | 1 | 1 | 0 | 8 |
| MRCP et al.,(2015) [7] | 1 | 0 | 1 | 0 | 1 | 1 | 1 | 0 | 0 | 5 |
| Li et al.,(2018A) [8] | 1 | 0 | 1 | 1 | 1 | 1 | 1 | 1 | 0 | 8 |
| Foureur et al.,(2001) [9] | 1 | 1 | 1 | 1 | 1 | 0 | 1 | 1 | 0 | 7 |
| Casas-de-la-Asuncinet al.,(2014) [10] | 1 | 0 | 1 | 1 | 1 | 1 | 1 | 1 | 0 | 7 |
| Brick, et al.,(2014) [11] | 1 | 1 | 1 | 1 | 1 | 1 | 1 | 1 | 0 | 8 |
| Langan et al.,(2011) [12] | 1 | 1 | 1 | 1 | 1 | 1 | 1 | 1 | 0 | 8 |
| Bastuji-Garin et al.,(2011) [13] | 1 | 0 | 1 | 1 | 1 | 1 | 1 | 1 | 0 | 7 |

dementia. The results were consistent, and no single study changed the significance of the pooled OR (Fig 3B). The funnel plot was symmetrical (Fig 4B), and Egger's test did not detect any funnel plot asymmetry (P = 0.085).

We conducted subgroup analysis and meta-regression analyses to examine the effect of different factors on the association between BP and all-cause dementia. Fig 5A shows the results of subgroup analysis by country or region. We found that Europe had a significantly higher odds ratio for case-control studies comparing BP and dementia (OR = 4.87,95%CI:3.27–7.25; P = 0.000) than Asia (OR = 1.88, 95%CI: 0.74–4.75; P = 0.184). Fig 5B shows the results of subgroup analysis by BP diagnosis method. Both subgroups of BP diagnosed by ICD codes (OR = 3.42,95%CI:1.26–9.31; P = 0.016) and BP diagnosed by clinical, histological, and immunofluorescence criteria (OR = 4.45,95%CI:3.12–6.33; P = 0.000) had significantly higher ORs for case-control studies comparing BP and all-cause dementia. Fig 5C shows the results of subgroup analysis by dementia diagnosis method. BP was significantly associated with all-cause dementia in subgroup of dementia diagnosed by medical records (OR = 4.81,95%CI:3.40–6.80; P = 0.000), but not in subgroup of dementia diagnosed by ICD codes (OR = 1.31,95%CI:0.88–1.96). Fig 5D shows the results of subgroup analysis by adjustment status. Both

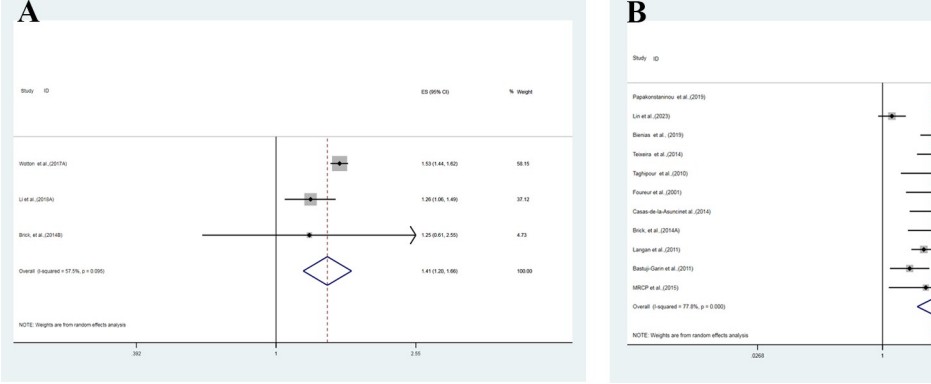

**Fig 2.** The forest plot shows the pooled HRs for longitudinal cohort(2A) and pooled ORs case-control (2B) association analysis between BP and all-cause dementia in middle-aged and older participants.

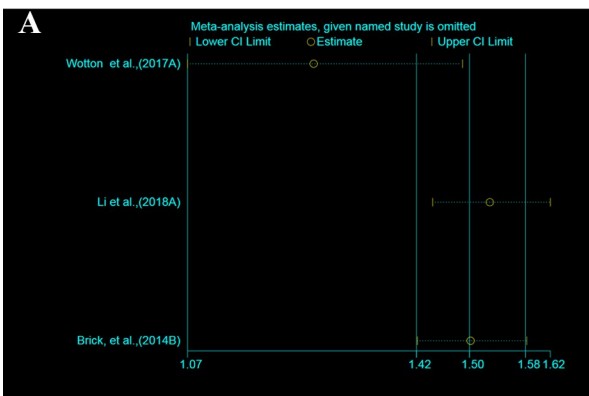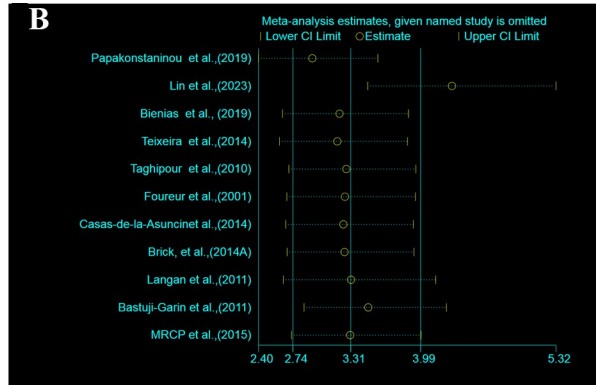

**Fig 3.** The sensitivity analyses performed to validate the stability of pooled HRs or ORs of literature by removing each individual study for for longitudinal cohort(3A) and case-control (3B) association analysis.

subgroups of adjusted and unadjusted studies had significantly higher ORs for case-control studies comparing BP and all-cause dementia (OR = 7.69,95%CI:4.88–12.13 and 3.48,95% CI:2.19–5.54; P = 0.000). The meta-regression analyses did not identify any other significant effect modifiers.

## Discussion

In this meta-analysis, we found that BP was associated with higher odds of all-cause dementia in middle-aged and older participants in both case-control and cohort studies. This finding is consistent with previous studies that reported an increased risk of dementia in patients with BP [1,2,8]. The possible mechanisms underlying this association are not fully understood, but some hypotheses have been proposed. One hypothesis is that BP and dementia share common risk factors, such as aging, genetic susceptibility, inflammation, oxidative stress, and vascular damage [10,15]. Another hypothesis is that BP may be a marker of neurodegeneration, as BP autoantibodies can cross-react with neuronal antigens and cause neuronal damage [31,32]. A third hypothesis is that BP may be a consequence of dementia, as cognitive impairment may impair skin barrier function and increase susceptibility to infections and trauma [6,33]. However, we did not find a significant difference in BP association between Alzheimer's dementia

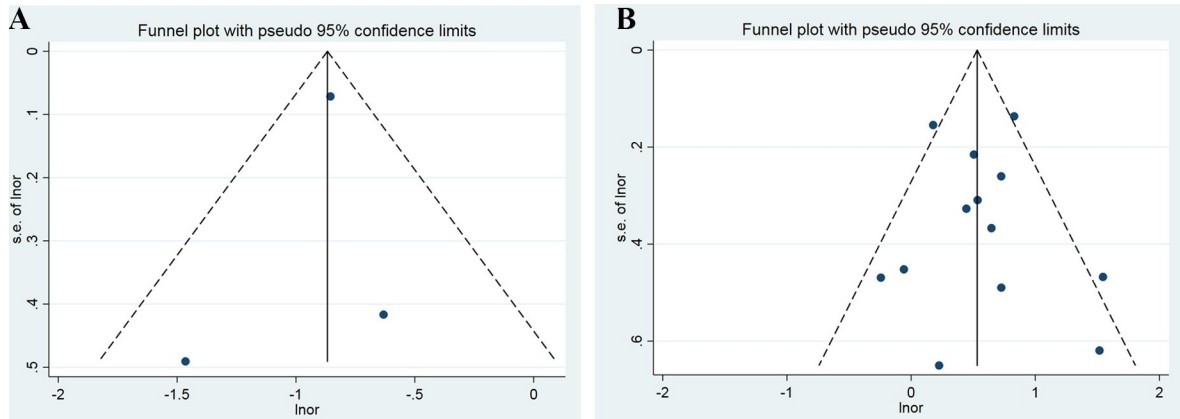

**Fig 4.** Funnel plots of studies evaluating the relationship between BP and all-cause dementia for longitudinal cohort(4A) and case-control (4B) association analysis in middle-aged and older participants.

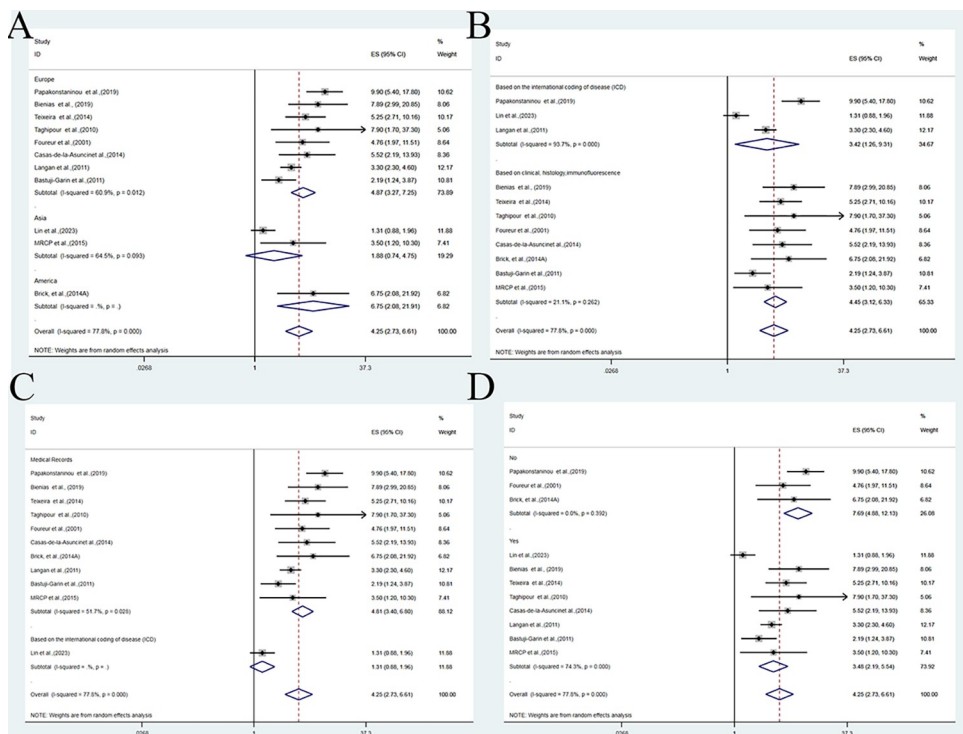

**Fig 5. Subgroup analysis and meta-regression analyses to examine the effect of different factors on the association between BP and all-cause dementia.** A: The forest plot of subgroup analysis by country or region. B: The forest plot of subgroup analysis by BP diagnosis method. C: The forest plot of subgroup analysis by medical records. D: The forest plot of subgroup analysis by adjustment status.

(AD) and vascular dementia (VaD) subtypes. This may be explained by the fact that both AD and VaD share common pathological features, such as amyloid plaques, neurofibrillary tangles, cerebral amyloid angiopathy, and white matter lesions [34,35]. Moreover, both AD and VaD have similar risk factors, such as age, diabetes, smoking, and hyperlipidemia [36,37]. Therefore, it is possible that BP has a similar impact on both types of dementia, regardless of the underlying etiology.

Our meta-analysis also identified some subgroups associated with significantly reported odds ratios, including Europe, BP diagnosed based on clinical, histology, immunofluorescence, and adjustment status of both NO and YES. For example, we found that the association between BP and dementia was stronger in studies conducted in Europe than Asia region. This may be due to the differences in genetic, environmental, and lifestyle factors between the two ethnic groups [38,39]. For example, European populations may have a higher intake of salt and alcohol, which are dietary factors that can increase BP and impair cognition [40,41]. Moreover, the subgroup analysis by dementia diagnosis method showed that BP was significantly associated with all-cause dementia in subgroup of dementia diagnosed by medical records, but not in subgroup of dementia diagnosed by ICD codes. This may be due to the possible misclassification and underdiagnosis of dementia using ICD codes, which may underestimate the true association between BP and dementia. Finally,we did not find significant differences in the association between BP and dementia by BP diagnosis method. Both subgroups of BP diagnosed by ICD codes and BP diagnosed by clinical, histological, and immunofluorescence criteria had significantly higher ORs for case-control studies comparing BP and all-cause dementia, suggesting that the association was not affected by the diagnostic accuracy of

BP. These results may also reflect differences in study design, population characteristics, diagnostic criteria, and confounding factors among the included studies. Therefore, caution is needed when interpreting and generalizing these results.

Our study has several implications for future research and clinical practice. First, our study suggests that BP management is important for preventing or delaying cognitive decline and dementia in older adults. Therefore, clinicians should monitor and treat hypertension in older adults according to the current guidelines [42]. Second, our study indicates that there may be ethnic differences in the BP-cognition relationship. Therefore, researchers should conduct more cross-cultural studies to explore the potential mechanisms and moderators of this relationship. Third, our study highlights the need for more studies to examine the differential effects of BP on different types of dementia. Therefore, researchers should use more accurate and comprehensive methods to diagnose and classify dementia subtypes in older adults.

Our meta-analysis has several strengths and limitations. The strengths include a comprehensive literature search, a rigorous study selection process, a quantitative synthesis of data from different study designs, and a thorough assessment of publication bias and heterogeneity. The limitations include the possibility of residual confounding, selection bias, misclassification bias and reverse causation due to the observational nature of the included studies; the variability in BP and dementia definitions, measurements, and assessments across studies; the lack of data on BP severity, duration and treatment; and the limited number of studies from some regions and subgroups.

## Conclusions

In conclusion, our meta-analysis suggests that BP is associated with higher odds of all-cause dementia in middle-aged and older participants. Further studies are needed to elucidate the causal mechanisms and clinical implications of this association.

## Supporting information

**S1 Table. PRISMA checklist.**
(DOCX)

**S2 Table. Search terms included for each library search.**
(DOCX)

**S3 Table. Overview of studies on the association between bullous pemphigoid and cognitive outcomes in included studies.**
(XLSX)

## Acknowledgments

We would like to thank Nanchang University for providing us with access to a series of available online databases.

## Author Contributions

**Conceptualization:** Qi Zhou, Xinming Li.

**Data curation:** Qi Zhou.

**Formal analysis:** Qi Zhou, Zhenrong Xiong, Dejiang Yang.

**Funding acquisition:** Qi Zhou.

**Investigation:** Qi Zhou, Dejiang Yang, Xinming Li.

**Methodology:** Qi Zhou, Dejiang Yang, Xinming Li.

**Project administration:** Chongyu Xiong.

**Resources:** Qi Zhou, Dejiang Yang, Chongyu Xiong, Xinming Li.

**Software:** Qi Zhou.

**Supervision:** Qi Zhou.

**Validation:** Qi Zhou, Dejiang Yang, Chongyu Xiong, Xinming Li.

**Visualization:** Qi Zhou, Zhenrong Xiong, Dejiang Yang, Xinming Li.

**Writing – original draft:** Qi Zhou, Xinming Li.

**Writing – review & editing:** Qi Zhou, Xinming Li.

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
