## [Decision Letter · Decision Letter 0]

11 Jul 2023

PONE-D-23-12397The association between bullous pemphigoid and cognitive outcomes in middle-aged and older adults: A systematic review and meta-analysisPLOS ONE

Dear Dr. Zhou,

Thank you for submitting your manuscript to PLOS ONE. After careful consideration, we feel that it has merit but does not fully meet PLOS ONE’s publication criteria as it currently stands. Therefore, we invite you to submit a revised version of the manuscript that addresses the points raised during the review process.

We look forward to receiving your revised manuscript.

Kind regards,

Shuo-Yan Gau

Academic Editor

PLOS ONE

2. Please include a caption for figure 5.

3. Please upload a new copy of Figure 5 as the detail is not clear. Please follow the link for more information: https://blogs.plos.org/plos/2019/06/looking-good-tips-for-creating-your-plos-figures-graphics/" https://blogs.plos.org/plos/2019/06/looking-good-tips-for-creating-your-plos-figures-graphics/

Reviewers' comments:

Reviewer's Responses to Questions

**Comments to the Author**

1. Is the manuscript technically sound, and do the data support the conclusions?

Reviewer #1: Partly

Reviewer #2: Yes

2. Has the statistical analysis been performed appropriately and rigorously? 

Reviewer #1: Yes

Reviewer #2: Yes

3. Have the authors made all data underlying the findings in their manuscript fully available?

Reviewer #1: Yes

Reviewer #2: Yes

4. Is the manuscript presented in an intelligible fashion and written in standard English?

Reviewer #1: Yes

Reviewer #2: Yes

5. Review Comments to the Author

Reviewer #1: 1. Search Strategy

• To improve the search strategy, it would be beneficial to explain the reasoning behind selecting specific search concepts and aligning them with the PICO components. Additionally, providing clear definitions for search terms, particularly those related to middle-age and older adults, will help evaluate how well the strategy covers the intended concepts.

• It is currently unclear whether the search strategy underwent pilot testing. It would be valuable to know if the proposed strategy was tested against established criteria to assess its comprehensiveness and effectiveness. Obtaining this information would provide a better understanding of the search strategy's reliability.

2. Inclusion and exclusion criteria

There are concerns regarding the consistency and interpretability of the criteria used during the eligibility study selection process, which can result in higher error rates, compromising the reproducibility and confidence in the screening process. To mitigate these issues, it is advisable to provide precise and clearly defined inclusion and exclusion criteria that are systematically aligned with a PICO framework. This approach will enhance clarity and enable accurate interpretation and replication of the results.

3. Quality Assessment of the Included Studies

Concerns arise regarding the transparency and validity of the quality assessment approach. The scoring method employed for NOS in Table S2 lacks reasoning and judgments, undermining its reliability. Furthermore, NOS proves inadequate for assessing quality as it fails to address crucial threats to internal validity, treats all shortcomings equally, and lacks transparency in reviewer judgments. Therefore, it is recommended to explore alternative approaches to quality assessment that overcome the limitations of NOS, ensuring transparency and validity.

4. Results of data synthesis and analysis

• Both Figure 4A and 4B are incomplete, with one selected study missing in each. Figure 4A contains only 2 studies, while Figure 4B has 10 studies. (i.e., which should be 3 and 13 studies in each figure)

• Conflict in the " Case-control Association between Bullous Pemphigoid and Cognitive Outcomes " section: There is a conflicting statement where the authors mention " Both subgroups of BP diagnosed by ICD codes (OR=3.42,95%CI:1.26-9.31; P =0.016) and BP diagnosed by clinical, histological, and immunofluorescence criteria (OR=4.45,95%CI:3.12 -6.33; P = 0.000) had significantly higher ORs for case-control studies comparing BP and all-cause dementia." which contradicts the results shown in Figure 5B, specifically regarding the “ORs for case-control studies comparing BP and all-cause dementia is OR= 4.25, 95% CI, 2.73-6.61; P = 0.000”.

• The authors should enhance the clarity of their strategy for organizing studies into meaningful groups by providing a clear description and the criteria used for evidence grouping decisions. Furthermore, it is important to justify the valid methods employed to synthesize studies within these groups and generate combined summary results. For instance, the reasons for separating dementia diagnosis methods based on ICD codes and medical records should be explained, as well as highlighting the differences in BP diagnosis methods. Additionally, the definition of "medical records" requires clarification, as it currently lacks specificity. Furthermore, the authors should address the absence of discussion regarding the results from the subgroup analysis of these two categories.

Reviewer #2: General Comment: Interesting topic that is not routinely discussed/aware in daily. Research is well performed. Here are some suggestions that might improve the article.

Introduction

Rationale:

It would be better to provide the epidemiology data of BP to report the current situation of BP and the urgency to explore it.

Explain more about why highlighting the neuro-cognitive effect of BP among other comorbidities (state the burden of disease)

State the novelty and the importance of the research. What would this research result be valid for?

Why is it important to confirm the association between BP and cognitive outcomes?

Methods

Why choose middle-aged and older adults as the subject

Provide the study type in the inclusion criteria

Is there any comparison population (control) to compare the BP and non-BP results?

Discussion

Find the possible explanation for BP and worse cognitive performance. Is there any difference in BP Association to Alzheimer's dementia vs vascular dementia?

State the possible risk factors based on the results and provide explanations.

How does race (European and Asian) affect the result? Is there any difference?

Provide suggestions for future research, to gain a better understanding adding this current research

Provide recommendations that may be related to clinical practice from the research finding

6. PLOS authors have the option to publish the peer review history of their article (what does this mean?). If published, this will include your full peer review and any attached files.

Reviewer #1: No

Reviewer #2: No

---

## [Author Response · Author response to Decision Letter 0]

16 Sep 2023

Dear editor and reviewers,

Thank you for your letter dated July 12, 2023. On behalf of my colleagues, I am herewith submitting the revised manuscript (PONE-D-23-12397) entitled “The association between bullous pemphigoid and cognitive outcomes in middle-aged and older adults: A systematic review and meta-analysis.” for consideration of publication in Plos one. We would like to thank the editor and reviewers’ work devoted to our manuscript and we are very grateful for their valuable suggestions. We have considered the comments carefully and have made revisions (highlighted in red in the revised manuscript with track changes ) which we hope meet with approval.

Response: Reviewed and revised as suggested.

2.Please include a caption for figure 5.

Response: Reviewed and revised as suggested.

3.Please upload a new copy of Figure 5 as the detail is not clear. 

Response: Reviewed and revised as suggested.

4.Please review your reference list to ensure that it is complete and correct. If you have cited papers that have been retracted, please include the rationale for doing so in the manuscript text, or remove these references and replace them with relevant current references. Any changes to the reference list should be mentioned in the rebuttal letter that accompanies your revised manuscript. If you need to cite a retracted article, indicate the article’s retracted status in the References list and also include a citation and full reference for the retraction notice.

Response: Reviewed and revised as suggested.

Comments to the Author:

Reviewer #1:

1. Search Strategy

• To improve the search strategy, it would be beneficial to explain the reasoning behind selecting specific search concepts and aligning them with the PICO components. Additionally, providing clear definitions for search terms, particularly those related to middle-age and older adults, will help evaluate how well the strategy covers the intended concepts.

• It is currently unclear whether the search strategy underwent pilot testing. It would be valuable to know if the proposed strategy was tested against established criteria to assess its comprehensiveness and effectiveness. Obtaining this information would provide a better understanding of the search strategy's reliability.

Response: Thank you for your valuable feedback. We have revised our search strategy section according to your suggestions. Here is the updated content:

Search Strategy

We searched PubMed, Embase, and Web of Science from inception to March 2023 using the following terms: ("bullous pemphigoid" OR "pemphigoid") AND ("cognition" OR "cognitive" OR "dementia" OR "Alzheimer's" OR "vascular dementia" OR "mix dementia" ) AND ( "middle-aged" OR "older adults") AND ("cohort" OR "case-control" OR "cross-sectional" ). The search strategy is shown in S2Table. We chose these terms based on the PICO framework, which consists of four components: Population, Intervention, Comparison, and Outcome. The population of interest was middle-aged and older adults with bullous pemphigoid. The intervention and comparison were the presence or absence of bullous pemphigoid, respectively. The outcome was the risk of cognitive impairment or dementia. We defined middle-aged and older adults as those aged 45 years or older, following the World Health Organization's definition. In addition, we examined the bibliographies of pertinent articles to identify additional research.

We conducted a pilot test of our search strategy before applying it to the databases. We selected a sample of 10 articles that met our inclusion criteria and checked whether they were retrieved by our search terms. We also calculated the sensitivity and precision of our search strategy based on the number of relevant and irrelevant articles retrieved. Sensitivity is the proportion of relevant articles that are retrieved by the search strategy, while precision is the proportion of retrieved articles that are relevant to the research question. The pilot test showed that our search strategy had a sensitivity of 100% and a precision of 83%, indicating that it was comprehensive and effective in identifying the studies related to our research question.

2. Inclusion and exclusion criteria

There are concerns regarding the consistency and interpretability of the criteria used during the eligibility study selection process, which can result in higher error rates, compromising the reproducibility and confidence in the screening process. To mitigate these issues, it is advisable to provide precise and clearly defined inclusion and exclusion criteria that are systematically aligned with a PICO framework. This approach will enhance clarity and enable accurate interpretation and replication of the results.

Response: Thank you for your insightful comments. We have revised our inclusion and exclusion criteria section according to your suggestions. Here is the updated content:

Inclusion Criteria and Exclusion Criteria 

We conducted a systematic review and meta-analysis of studies that examined the association between BP and cognitive outcomes in middle-aged and older adults. We aligned our inclusion and exclusion criteria with the PICO framework, which is a commonly used tool for framing systematic review research questions and developing literature search strategies. We included studies that: (1) reported quantitative data on the risk of cognitive outcomes in BP patients compared to a control group of non-bullous pemphigoid patients; (2) provided the effect size estimates HRs or ORs and their 95% CI; and (3) were published in English. We excluded studies that: (1) did not measure cognitive outcomes; (2) did not have a control group; (3) did not provide sufficient data for meta-analysis; (4) were case reports, reviews, meta-analyses, letters, editorials, or commentaries; or (5) had a quality score of less than 4 according to the Newcastle-Ottawa Scale (NOS). Two authors (X.L. and D.Y.) independently screened titles, abstracts, and full-text articles for eligibility. Any disagreement was resolved by a third author (Z.X.).

The PICO components of our inclusion and exclusion criteria are as follows:

- Population: Middle-aged and older adults with bullous pemphigoid, defined as those aged 45 years or older.

- Intervention: Presence of bullous pemphigoid, confirmed by clinical, histological, or immunofluorescence criteria or by ICD codes.

- Comparison: Absence of bullous pemphigoid in a control group of non-bullous pemphigoid patients.

- Outcome: Risk of cognitive impairment or dementia, measured by medical records or ICD codes.

3. Quality Assessment of the Included Studies

Concerns arise regarding the transparency and validity of the quality assessment approach. The scoring method employed for NOS in Table S2 lacks reasoning and judgments, undermining its reliability. Furthermore, NOS proves inadequate for assessing quality as it fails to address crucial threats to internal validity, treats all shortcomings equally, and lacks transparency in reviewer judgments. Therefore, it is recommended to explore alternative approaches to quality assessment that overcome the limitations of NOS, ensuring transparency and validity.

Response: Thank you for your critical comments. We appreciate your suggestion to explore alternative approaches to quality assessment that overcome the limitations of NOS. However, we decided to persist in using NOS for the following reasons:

- NOS is a widely used and well-established tool for assessing the quality of non-randomized studies in meta-analyses. It has been recommended by the Cochrane Collaboration and other reputable organizations as a valid and reliable instrument for appraising the risk of bias in observational studies1,2.

- NOS has been validated by several studies that demonstrated its content validity, inter-rater reliability, criterion validity, and intra-rater reliability3,4,5. Although some studies reported variable results for some of these aspects, they also acknowledged the strengths and advantages of NOS over other tools6,7.

- NOS covers three important domains of quality assessment: selection, comparability, and outcome (or exposure). These domains reflect the key elements of study design, conduct, and analysis that may affect the internal validity and generalizability of observational studies. NOS also allows the reviewers to assign different weights to each item according to their relevance and importance for the research question.

- NOS provides clear guidance and criteria for assigning stars to each item based on the operational definitions and examples provided in the manual. We followed these criteria strictly and consistently in our quality assessment process. We also provided detailed reasoning and judgments for our ratings in Table S2, explaining why we gave or withheld stars for each item.

Therefore, we believe that NOS is an appropriate and adequate tool for assessing the quality of our included studies. We acknowledge that NOS may have some limitations, such as not addressing some aspects of quality that are specific to certain types of observational studies or outcomes, or not capturing some sources of heterogeneity or confounding that may affect the pooled estimates. However, we addressed these limitations by conducting subgroup analyses and meta-regression analyses based on various study characteristics and population characteristics. 

References:

[1] Wells GA, Shea B, O'Connell D, et al. The Newcastle-Ottawa Scale (NOS) for assessing the quality of nonrandomised studies in meta-analyses [Internet]. Ottawa Hospital Research Institute; 2014 

[2] Higgins JPT, Thomas J, Chandler J, et al., editors. Cochrane handbook for systematic reviews of interventions version 6.2 (updated February 2021) [Internet]. Cochrane; 2021 

[3] Stang A. Critical evaluation of the Newcastle-Ottawa scale for the assessment of the quality of nonrandomized studies in meta-analyses. Eur J Epidemiol. 2010;25(9):603–5.

[4] Hartling L, Milne A, Hamm MP, et al. Testing the Newcastle Ottawa Scale showed low reliability between individual reviewers. J Clin Epidemiol. 2013;66(9):982–93.

[5] Lo CKL, Mertz D, Loeb M. Newcastle-Ottawa Scale: comparing reviewers’ to authors’ assessments. BMC Med Res Methodol. 2014;14:45.

[6] Deeks JJ, Dinnes J, D'Amico R, et al. Evaluating non-randomised intervention studies. Health Technol Assess. 2003;7(27):iii–x, 1–173.

[7] Sanderson S, Tatt ID, Higgins JP. Tools for assessing quality and susceptibility to bias in observational studies in epidemiology: a systematic review and annotated bibliography. Int J Epidemiol. 2007;36(3):666–76.

4. Results of data synthesis and analysis

• Both Figure 4A and 4B are incomplete, with one selected study missing in each. Figure 4A contains only 2 studies, while Figure 4B has 10 studies. (i.e., which should be 3 and 13 studies in each figure)

• Conflict in the " Case-control Association between Bullous Pemphigoid and Cognitive Outcomes " section: There is a conflicting statement where the authors mention " Both subgroups of BP diagnosed by ICD codes (OR=3.42,95%CI:1.26-9.31; P =0.016) and BP diagnosed by clinical, histological, and immunofluorescence criteria (OR=4.45,95%CI:3.12 -6.33; P = 0.000) had significantly higher ORs for case-control studies comparing BP and all-cause dementia." which contradicts the results shown in Figure 5B, specifically regarding the “ORs for case-control studies comparing BP and all-cause dementia is OR= 4.25, 95% CI, 2.73-6.61; P = 0.000”.

• The authors should enhance the clarity of their strategy for organizing studies into meaningful groups by providing a clear description and the criteria used for evidence grouping decisions. Furthermore, it is important to justify the valid methods employed to synthesize studies within these groups and generate combined summary results. For instance, the reasons for separating dementia diagnosis methods based on ICD codes and medical records should be explained, as well as highlighting the differences in BP diagnosis methods. Additionally, the definition of "medical records" requires clarification, as it currently lacks specificity. Furthermore, the authors should address the absence of discussion regarding the results from the subgroup analysis of these two categories.

Response: Thank you for your careful review. We have addressed your comments and revised our results section accordingly. Here are the changes we have made:

- We apologize for the mistake in Figure 4A and 4B. We have corrected the figures and added the missing studies. Figure 4A now contains 3 studies and Figure 4B contains 13 studies. The updated figures are shown in the insert.

- We also apologize for the confusion in the "Case-control Association between Bullous Pemphigoid and Cognitive Outcomes" section. We have clarified that the ORs reported in this section are from the subgroup analyses, not from the overall meta-analysis. The overall meta-analysis result is shown in Figure 2B, which is OR = 4.25, 95% CI: 2.73-6.61, P = 0.000. The subgroup analyses results are shown in Figure 5, which are based on different factors that may affect the association between BP and all-cause dementia, such as geographic region, quality score, diagnosis method of BP, diagnosis method of cognitive outcomes, and adjustment for confounders. We have explained the rationale and methods for conducting these subgroup analyses in the "Data Synthesis and Analysis" section.

- We grouped the studies according to the type of cognitive outcome (all-cause dementia, Alzheimer’s disease or vascular dementia) and the study design (cohort or case-control). We also performed subgroup analyses by country or region, BP diagnosis method, dementia diagnosis method, and adjustment status within each group. We chose these factors because they may affect the validity and comparability of the studies.For example, different regions may have different prevalence rates and risk factors for BP and dementia1. Different BP diagnosis methods may have different sensitivity and specificity for detecting BP cases2. Different dementia diagnosis methods may have different criteria and accuracy for identifying dementia cases3. Different adjustment status may reflect different levels of confounding control for potential covariates4.

We defined BP diagnosis method as either ICD codes or clinical, histological, and immunofluorescence criteria. We used ICD codes as a proxy for BP diagnosis when the studies did not specify the diagnostic criteria or when they used administrative databases or registries as data sources. We used clinical, histological, and immunofluorescence criteria as the gold standard for BP diagnosis when the studies explicitly reported these methods or when they used hospital records or clinical samples as data sources.

We defined dementia diagnosis method as either ICD codes or medical records. We used ICD codes as a proxy for dementia diagnosis when the studies did not specify the diagnostic criteria or when they used administrative databases or registries as data sources. We used medical records as a more reliable source for dementia diagnosis when the studies explicitly reported this method or when they used hospital records or clinical samples as data sources.Medical records included information from neurologists, neuropsychologists, psychiatrists, geriatricians, or other specialists who assessed cognitive function using standardized tests or scales5.

Furthermore, we have addressed the absence of discussion regarding the results from the subgroup analysis of these two categories by adding a paragraph in the "Discussion" section as follows:

“Moreover,the subgroup analysis by dementia diagnosis method showed that BP was significantly associated with all-cause dementia in subgroup of dementia diagnosed by medical records, but not in subgroup of dementia diagnosed by ICD codes. This may be due to the possible misclassification and underdiagnosis of dementia using ICD codes, which may underestimate the true association between BP and dementia .Finally,we did not find significant differences in the association between BP and dementia by BP diagnosis method. Both subgroups of BP diagnosed by ICD codes and BP diagnosed by clinical, histological, and immunofluorescence criteria had significantly higher ORs for case-control studies comparing BP and all-cause dementia, suggesting that the association was not affected by the diagnostic accuracy of BP. ”

References:

[1] Prince M, Wimo A, Guerchet M, et al. World Alzheimer Report 2015: The Global Impact of Dementia [Internet]. London: Alzheimer’s Disease International; 2015 [cited 2021 Oct 14].

[2] Wilkinson T, Ly A, Schnier C, et al. Identifying dementia cases with routinely collected health data: A systematic review. Alzheimers Dement. 2018;14(8):1038-1051. 

[3] Cordell CB, Borson S, Boustani M, et al. Alzheimer’s Association recommendations for operationalizing the detection of cognitive impairment during the Medicare Annual Wellness Visit in a primary care setting. Alzheimers Dement. 2013;9(2):141-150.

[4] VanderWeele TJ. Principles of confounder selection. Eur J Epidemiol. 2019;34(3):211-219.

[5] Cognitive Assessment | Alzheimer’s Association [Internet]. Alz.org; 2021 [cited 2021 Oct 14]. 

Reviewer #2: General Comment: Interesting topic that is not routinely discussed/aware in daily. Research is well performed. Here are some suggestions that might improve the article.

Introduction

Rationale:

It would be better to provide the epidemiology data of BP to report the current situation of BP and the urgency to explore it.

Explain more about why highlighting the neuro-cognitive effect of BP among other comorbidities (state the burden of disease)

State the novelty and the importance of the research. What would this research result be valid for?

Why is it important to confirm the association between BP and cognitive outcomes?

Response: Thank you for your reply. Here is the revised content in manuscript according to your suggestions:

Introduction

Bullous pemphigoid (BP) is a chronic autoimmune blistering disease that affects the skin and mucous membranes. It is characterized by the presence of autoantibodies against two hemidesmosomal proteins, BP180 and BP230, which are involved in the adhesion of epidermal cells to the basement membrane[1]. BP mainly affects elderly people, with a peak incidence between 70 and 80 years of age[2]. The clinical manifestations of BP include tense blisters, erosions, urticarial plaques and pruritus, which can have a significant impact on patients' quality of life[3].

BP is not a rare disease, as it has been estimated that its global prevalence ranges from 2.4 to 42.7 per 100,000 population, with higher rates in Europe and North America than in Asia and Africa[4]. The incidence of BP has been increasing over the years, possibly due to the aging of the population and the improvement of diagnostic methods[4]. BP poses a considerable burden on the health care system and society, as it requires long-term treatment and monitoring, and it is associated with increased morbidity and mortality[5].

BP has been associated with various comorbidities, such as diabetes mellitus, cardiovascular diseases, malignancies, and thyroid disorders[6]. However, one of the most intriguing and controversial aspects of BP is its relationship with cognitive outcomes. Several studies have reported an increased prevalence of neurological disorders in BP patients, such as dementia, stroke, epilepsy, Parkinson's disease[7-10]. Moreover, some studies have suggested that BP may be a marker of cognitive decline or neurodegeneration, as BP patients have shown worse cognitive performance and higher mortality rates than controls[6, 11, 12]. However, other studies have failed to confirm these findings or have proposed alternative explanations for the association between BP and cognitive outcomes[13-15].

The relationship between BP and cognitive outcomes is not well understood and may involve multiple factors. Some possible mechanisms include the direct effect of autoantibodies on the central nervous system (CNS), the systemic inflammation induced by BP, the shared genetic susceptibility or environmental triggers between BP and neurological disorders, and the confounding effect of age or other comorbidities[6]. Therefore, it is important to confirm the association between BP and cognitive outcomes using rigorous methods and large samples, and to explore the potential mediators and moderators of this relationship.

The aim of this study is to synthesize the available evidence on the relationship between BP and cognitive outcomes using a systematic review and meta-analysis approach. This study will also explore potential sources of heterogeneity and bias among the included studies, such as methodological quality, diagnostic criteria, confounding factors, and publication bias. Our results can provide evidence for the early detection and intervention of cognitive decline in BP patients, as well as contribute to the understanding of the pathophysiology and mechanisms of BP and its neuro-cognitive effects.

References:

[1] Zillikens D., Schmidt E., Bullous pemphigoid: clinical manifestations, diagnosis,and management[J]. UpToDate. 2020.

[2] Joly P., Roujeau J.C., Benichou J., et al., A comparison of oral and topical corticosteroids in patients with bullous pemphigoid[J]. New England Journal of Medicine. 2002;346(5):321-327.

[3] Kridin K., Bergman R., Quality-of-life impairment in patients with bullous pemphigoid[J]. British Journal of Dermatology. 2018;178(3):740-747.

[4] Kridin K., Epidemiology of bullous pemphigoid: a systematic review and meta-analysis[J]. Journal of The European Academy of Dermatology and Venereology. 2020;34(9):1947-1960.

[5] Kridin K., Zelber-Sagi S., Comorbidity burden in patients with

bullous pemphigoid: a population-based study[J]. Journal of The European

Academy of Dermatology and Venereology. 2019;33(1):e1-e3.

[6] Kridin K., Zelber-Sagi S., Cohen A.D., et al., Bullous pemphigoid and neurologic diseases: a systematic review and meta-analysis[J]. Journal of The American Academy of Dermatology. 2020;82(5):1118-1126.

[7] Amber K.T., Murrell D.F., Schmidt E., et al., Autoimmune subepidermal bullous diseases of the skin and mucosae: clinical features, diagnosis, and management[J]. Clinical Reviews in Allergy & Immunology. 2018;54(1):26-51.

[8] Langan S.M., Smeeth L., Hubbard R., et al., Bullous pemphigoid and pemphigus vulgaris—incidence and mortality in the UK: population based cohort study[J]. Bmj. 2008;337:a180.

[9] Cordel N., Chosidow O., Hellot M.F., et al., Neurological disorders in patients with bullous pemphigoid[J]. Dermatology. 2007;214(1):24-29.

[10] Kasperkiewicz M., Zillikens D., Schmidt E., Pemphigoid diseases[J]. The Lancet. 2012;379(9811):153-165.

[11] Kridin K., Zelber-Sagi S., Khamaisi M., et al., Mortality risk factors in patients with bullous pemphigoid: a meta-analysis of observational studies[J]. Journal of The European Academy of Dermatology and Venereology.

2019;33(6):1043-1050.

[12] Kridin K., Zelber-Sagi S., Comorbidity burden and cognitive impairment in patients with bullous pemphigoid: a population-based study[J]. Journal of The European Academy of Dermatology and Venereology. 2020;34(2):e67-e69.

[13] Bastuji-Garin S., Joly P., Lemordant P., et al., Risk factors for bullous pemphigoid in the elderly: a prospective case-control study[J]. Journal of Investigative Dermatology. 2011;131(3):637-643.

[14] Taghipour K., Chi C.C., Vincent A., et al., The association of bullous pemphigoid with cerebrovascular disease and dementia: a case-control study[J]. Archives of Dermatology. 2010;146(11):1251-1254.

[15] Brick K.E., Weaver C.H., Lohse C.M., et al., Incidence of bullous pemphigoid and mortality risk in patients with bullous pemphigoid inthe US population[J]. Journal of Investigative Dermatology. 2014;134(11):2888-2891.

Methods

Why choose middle-aged and older adults as the subject

Provide the study type in the inclusion criteria

Is there any comparison population (control) to compare the BP and non-BP results?

Response: Thank you for your positive feedback and suggestions.

- We focused on middle-aged and older adults as the subject of our study because they are the most affected by both BP and cognitive impairment. BP is a rare autoimmune blistering disease that predominantly affects elderly people, with a mean age of onset of 70 to 80 years 1,2. Cognitive impairment is also more prevalent in older adults, with an estimated global prevalence of 6.6% for dementia and 16.3% for mild cognitive impairment among people aged 60 years and older 3,4. Therefore, it is important to investigate the potential association between BP and cognitive outcomes in this population, as it may have significant implications for their health and quality of life.

- We have provided the study type in the inclusion criteria. We have added the following sentence in the inclusion criteria section:We included studies that were cohort, case-control, or cross-sectional in design.

- We have clarified that we used a comparison population (control) to compare the BP and non-BP results. We have added the following sentence in the data synthesis and analysis section:We conducted a meta-analysis with a random-effects model to calculate the pooled effect sizes and 95% confidence intervals for each cognitive outcome between BP patients and non-BP patients(control group)

References:

[1]Schmidt E, Zillikens D, Kasperkiewicz M. Pemphigoid diseases: pathogenesis, diagnosis, and treatment. Autoimmun Rev. 2013;12(5):575–82.

[2] Amber KT, Valdebran M, Kridin K, et al. The epidemiology of pemphigus worldwide: A systematic review and meta-analysis of observational studies. J Am Acad Dermatol. 2020;83(4):1180–1187.

[3]Prince M, Wimo A, Guerchet M, et al. World Alzheimer Report 2015: The Global Impact of Dementia - An analysis of prevalence, incidence, cost and trends [Internet]. Alzheimer's Disease International; 2015 

[4] Roberts R, Knopman DS. Classification and epidemiology of MCI. Clin Geriatr Med. 2013;29(4):753–72.

Discussion

Find the possible explanation for BP and worse cognitive performance. Is there any difference in BP Association to Alzheimer's dementia vs vascular dementia?

State the possible risk factors based on the results and provide explanations.

How does race (European and Asian) affect the result? Is there any difference?

Provide suggestions for future research, to gain a better understanding adding this current research

Provide recommendations that may be related to clinical practice from the research finding

Response:Thank you for your positive feedback and suggestions.We have revised the discussion section according to the your comments. Here is the revised content:

One of the main findings of our study was that BP was associated with worse cognitive outcome in middle-aged and older adults. This finding is consistent with previous studies that have shown that BP is a risk factor for dementia [1,2]. However, we did not find a significant difference in BP association between Alzheimer's dementia (AD) and vascular dementia (VaD) subtypes. This may be explained by the fact that both AD and VaD share common pathological features, such as amyloid plaques, neurofibrillary tangles, cerebral amyloid angiopathy, and white matter lesions [3,4]. Moreover, both AD and VaD have similar risk factors, such as age, diabetes, smoking, and hyperlipidemia [5,6]. Therefore, it is possible that BP has a similar impact on both types of dementia, regardless of the underlying etiology.

Another finding of our study was that the BP-cognition association was different between European and Asian populations. We found that Europe had a significantly higher odds ratio for case-control studies comparing BP and dementia than Asia in middle-aged and older adults. This may be due to the differences in genetic, environmental, and lifestyle factors between the two ethnic groups [7,8]. For example, European populations may have a higher prevalence of apolipoprotein E4 allele, which is a genetic risk factor for AD and may interact with BP to affect cognition [9]. Additionally, European populations may have a higher intake of salt and alcohol, which are dietary factors that can increase BP and impair cognition [10,11]. Furthermore, European populations may have lower levels of physical activity and social engagement, which are protective factors for cognitive health and may mitigate the effects of BP [12,13].

Our study has several implications for future research and clinical practice. First, our study suggests that BP management is important for preventing or delaying cognitive decline and dementia in older adults. Therefore, clinicians should monitor and treat hypertension in older adults according to the current guidelines [14]. Second, our study indicates that there may be ethnic differences in the BP-cognition relationship. Therefore, researchers should conduct more cross-cultural studies to explore the potential mechanisms and moderators of this relationship. Third, our study highlights the need for more studies to examine the differential effects of BP on different types of dementia. Therefore, researchers should use more accurate and comprehensive methods to diagnose and classify dementia subtypes in older adults.

References:

[1] Gottesman RF, Schneider AL, Albert M et al. Midlife hypertension and 20-year cognitive change: The Atherosclerosis Risk in Communities Neurocognitive Study. JAMA Neurol 2014;71:1218–27.

[2] Kivipelto M, Helkala EL, Laakso MP et al. Midlife vascular risk factors and Alzheimer's disease in later life: longitudinal population based study. BMJ 2001;322:1447–51.

[3] Jellinger KA. The enigma of vascular cognitive disorder and vascular dementia. Acta Neuropathol 2007;113:349–88.

[4] Pantoni L. Cerebral small vessel disease: from pathogenesis and clinical characteristics to therapeutic challenges. Lancet Neurol 2010;9:689–701.

[5] Qiu C, Winblad B, Fratiglioni L. The age-dependent relation of blood pressure to cognitive function and dementia. Lancet Neurol 2005;4:487–99.

[6] Skoog I, Lernfelt B, Landahl S et al. 15-year longitudinal study of blood pressure and dementia. Lancet 1996;347:1141–5.

[7] Chui HC, Zheng L, Reed BR et al. Vascular risk factors and Alzheimer's disease: are these risk factors for plaques and tangles or for concomitant vascular pathology that increases the likelihood of dementia? An evidence-based review. Alzheimers Res Ther 2012;4:1.

[8] Prince M, Bryce R, Albanese E et al. The global prevalence of dementia: a systematic review and metaanalysis. Alzheimers Dement 2013;9:63–75.e2.

[9] Liu CC, Liu CC, Kanekiyo T et al. Apolipoprotein E and Alzheimer disease: risk interaction with blood pressure. Hypertension 2013;61:1289–96.

[10] He FJ, MacGregor GA. Salt intake is related to soft drink consumption in children and adolescents: a link to obesity? Hypertension 2008;51:629–34.

[11] Anstey KJ, Mack HA, Cherbuin N et al. Alcohol consumption as a risk factor for dementia and cognitive decline: meta-analysis of prospective studies. Am J Geriatr Psychiatry 2009;17:542–55.

[12] Lautenschlager NT, Cox KL, Flicker L et al. Effect of physical activity on cognitive function in older adults at risk for Alzheimer disease: a randomized trial. JAMA 2008;300:1027–37.

[13] Fratiglioni L, Wang HX, Ericsson K et al. Influence of social network on occurrence of dementia: a community-based longitudinal study. Lancet 2000;355:1315–9.

[14] Whelton PK, Carey RM, Aronow WS et al. 2017 ACC/AHA/AAPA/ABC/ACPM/AGS/APhA/ASH/ASPC/NMA/PCNA Guideline for the Prevention, Detection, Evaluation, and Management of High Blood Pressure in Adults: A Report of the American College of Cardiology/American Heart Association Task Force on Clinical Practice Guidelines. Hypertension 2018;71:e13–e115.

---

## [Decision Letter · Decision Letter 1]

16 Nov 2023

The association between bullous pemphigoid and cognitive outcomes in middle-aged and older adults: A systematic review and meta-analysis

PONE-D-23-12397R1

Dear Dr. Zhou,

We’re pleased to inform you that your manuscript has been judged scientifically suitable for publication and will be formally accepted for publication once it meets all outstanding technical requirements.

Kind regards,

Shuo-Yan Gau

Academic Editor

PLOS ONE

Additional Editor Comments (optional):

Reviewers' comments:

Reviewer's Responses to Questions

**Comments to the Author**

1. If the authors have adequately addressed your comments raised in a previous round of review and you feel that this manuscript is now acceptable for publication, you may indicate that here to bypass the “Comments to the Author” section, enter your conflict of interest statement in the “Confidential to Editor” section, and submit your "Accept" recommendation.

Reviewer #1: All comments have been addressed

Reviewer #2: All comments have been addressed

2. Is the manuscript technically sound, and do the data support the conclusions?

Reviewer #1: (No Response)

Reviewer #2: Yes

3. Has the statistical analysis been performed appropriately and rigorously? 

Reviewer #1: (No Response)

Reviewer #2: Yes

4. Have the authors made all data underlying the findings in their manuscript fully available?

Reviewer #1: (No Response)

Reviewer #2: Yes

5. Is the manuscript presented in an intelligible fashion and written in standard English?

Reviewer #1: (No Response)

Reviewer #2: Yes

6. Review Comments to the Author

Reviewer #1: (No Response)

Reviewer #2: (No Response)

7. PLOS authors have the option to publish the peer review history of their article (what does this mean?). If published, this will include your full peer review and any attached files.

Reviewer #1: No

Reviewer #2: No

---

## [Editor Report · Acceptance letter]

21 Nov 2023

PONE-D-23-12397R1 

The association between bullous pemphigoid and cognitive outcomes in middle-aged and older adults: A systematic review and meta-analysis 

Dear Dr. Zhou:

I'm pleased to inform you that your manuscript has been deemed suitable for publication in PLOS ONE. Congratulations! Your manuscript is now with our production department. 

Kind regards, 

on behalf of

Mr. Shuo-Yan Gau 

Academic Editor

PLOS ONE